# Seroconversion of a Swine Herd in a Free-Range Rural Multi-Species Farm against HPAI H5N1 2.3.4.4b Clade Virus

**DOI:** 10.3390/microorganisms11051162

**Published:** 2023-04-28

**Authors:** Francesca Rosone, Francesco Bonfante, Marcello Giovanni Sala, Silvia Maniero, Antonella Cersini, Ida Ricci, Luisa Garofalo, Daniela Caciolo, Antonella Denisi, Alessandra Napolitan, Monja Parente, Bianca Zecchin, Calogero Terregino, Maria Teresa Scicluna

**Affiliations:** 1Istituto Zooprofilattico Sperimentale del Lazio e della Toscana “M. Aleandri”, Via Appia Nuova, 1411, 00178 Rome, Italy; marcello.sala@izslt.it (M.G.S.); antonella.cersini@izslt.it (A.C.); ida.ricci@izslt.it (I.R.); luisa.garofalo@izslt.it (L.G.); daniela.caciolo@izslt.it (D.C.); antonella.denisi@izslt.it (A.D.); teresa.scicluna@izslt.it (M.T.S.); 2Istituto Zooprofilattico Sperimentale delle Venezie (IZSVe), 35020 Legnaro, Italy; fbonfante@izsve.it (F.B.); smaniero@izsve.it (S.M.); anapolitan@izsve.it (A.N.); bzecchin@izsve.it (B.Z.); 3State Veterinarians of the Local Health Unit (LHU), 00054 Rome, Italy; monja.parente@aslroma3.it

**Keywords:** HPAI H5N1, free-ranging multispecies farm, poultry, swine

## Abstract

Starting from October 2021, several outbreaks of highly pathogenic avian influenza virus (HPAIV) subtype H5N1 were reported in wild and domestic birds in Italy. Following the detection of an HPAIV in a free-ranging poultry farm in Ostia, province of Rome, despite the lack of clinical signs, additional virological and serological analyses were conducted on samples collected from free-ranging pigs, reared in the same holding, due to their direct contact with the infected poultry. While the swine nasal swabs were all RT-PCR negative for the influenza type A matrix (M) gene, the majority (%) of the tested pigs resulted serologically positive for the hemagglutination inhibition test and microneutralization assay, using an H5N1 strain considered to be homologous to the virus detected in the farm. These results provide further evidence of the worrisome replicative fitness that HPAI H5Nx viruses of the 2.3.4.4b clade have in mammalian species. Moreover, our report calls for additional active surveillance, to promptly intercept occasional spillover transmissions to domestic mammals in close contact with HPAI affected birds. Strengthened biosecurity measures and efficient separation should be prioritized in mixed-species farms in areas at risk of HPAI introduction.

## 1. Introduction

During 2021–2022, one of the most severe waves of highly pathogenic avian influenza (HPAI) virus to hit the European continent, affecting wild, captive and domestic birds with heavy repercussions on the production and export of poultry products, and with a severe impact on the operational capacity of state veterinarians, regional laboratories, public health authorities and local practitioners. During this epidemic wave, the absolute majority of the HPAI viruses detected in Europe were of the H5N1 subtype and belonged to the clade 2.3.4.4b of the Goose/Guangdong (Gs/GD) lineage.

Phylogeographic analyses indicated the presence of at least six genotypes of H5N1, most likely as the direct result of an extensive circulation of these viruses in the wild bird reservoir and the frequent reassortment events with other subtypes of avian influenza viruses [1]. Interestingly, the high prevalence of infection in wild birds was consistently associated with unprecedented high numbers of mammalian HPAI infections in Europe and North America, where over 24 species of predators and scavengers were found either dead or showing severe signs of neurological impairment, referable to avian influenza infection [2]. Viruses of the 2.3.4.4b clade are actually the most prevalent at a global level, affecting not only Europe and large areas of the Asian continent, but also Africa, North America and are known to have recently spread for the first time to Central and South America [3,4]. 

From October 2021 to January 2022, over 300 outbreaks caused by H5N1 subtype HPAI viruses of the clade 2.3.4.4b were detected in various Italian Regions, including the Latium Region, where two cases of HPAI were diagnosed, namely in a family-run mixed-species backyard farm and in wild birds found sick in a small lake of a municipal villa, both in the province of Rome. It is known that farming settings where several avian species are reared together offer an opportunity for AI viruses to perpetuate more or less silently, increasing the likelihood of reassortment events. Moreover, rural settings where avian and mammalian domestic species are bred in close contact are at risk of not only interspecies transmission, but also of generating reassortant influenza viruses of pandemic concern through replication and adaptation to mixing vessels such as pigs [5].

Type A swine influenza viruses of the H1 and H3 subtypes are enzootic in pigs and originated either directly from birds, (e.g., the Eurasian H1N1 lineage), through reverse-spillover events (e.g., pandemic H1N1 viruses) or following reassortment events between human and swine viruses (e.g., the H3N2 influenza A variant viruses) [6]. Interspecies transmission of avian influenza virus (AIV) to pigs was reported in Canada in 1999 for a low pathogenic H4N6 virus, followed by reports of an H3N3 and H1N1 strain in Ontario in 2001 [7]. So far, the large majority of AIV transmissions to swine have occurred in China, including low-pathogenicity AIV of the H9N2, H7N9, H1N5, H4N1, H6N6 and H4N8 subtypes [8,9]. In addition, H5N2, H7N2 and H9N2 viruses were reported in South Korean pigs in 2001, 2004 and 2008, respectively [10]. Genomes of highly pathogenic H5N1 viruses were also reported in Indonesian pigs in the 2005–2007 period. More recently, in 2018, the genome of an H5N2 virus was reported in Mexican pigs [11]. Most importantly, during the 2016–2017 season, serosurveillance of ten pig herds in southwest France revealed that in one of the herds, one pig had seroconverted by the hemagglutination inhibition assay to a 2.3.4.4.b clade virus [12]. On the basis of these data, pigs appear to be susceptible to infections with both LPAI and HPAI viruses, although replication in these animals is characterized by poor or negligible clinical manifestations and onward transmission [13], as demonstrated by the limited number of detections and by the consistent association between infection and direct exposure to infected poultry [14].

Although there are a number of papers describing natural H5N1 HPAI virus infection of animals taxonomically belonging to the order Carnivora (i.e., domestic cats, tigers, leopards, dogs and stone martens), data on isolation of H5N1 viruses from pigs (*Sus scrofa*, family Suidae, order Artiodactyla) are limited, in both natural and experimental infection [15]. Challenge studies done in the past confirm that pigs could be infected with avian IAV, but in most cases they do not show any clinical signs. It appears that the virus replicates only in the lower part of the lung where avian AIV receptors (alpha-2,3 sialic acids) are present. However, the virus is transmitted poorly or not at all between pigs or from pigs to other mammals, humans included [16,17]. Furthermore, the susceptibility of pigs to avian influenza viruses (AIV) is dependent on many variables such as virus subtype, infectious dose and/or route of inoculation. 

The present paper describes the serological, virological and epidemiological investigations conducted in a mixed-species farm with pigs and other domestic birds in the province of Rome, where an outbreak of H5N1 HPAI virus was diagnosed in October 2021.

## 2. Background and Event History

From 27 October to 2 November 2021, state veterinarians recorded the sudden death of 99% (60/61) of the laying hens reared in a small backyard farm (Figure 1) in the municipality of Ostia, in the province of Rome, Italy.

Of the sixty carcasses, specimens (intestine, spleen, lungs and air sacs) from only three dead laying hens were collected by veterinarians of Rome 3 Local Health Unit (LHU) for immediate diagnostic purposes.

The samples were examined at the Virology Unit of the Istituto Zooprofilattico Sperimentale delle Regioni Lazio e Toscana and were found positive for AIV by real-time RT-PCR (RRT-PCR) for the Matrix (M) gene as described below. Following this, the remaining carcasses were immediately sent for destruction. The virus was subsequently sent to the National Reference Center for Avian Influenza and Newcastle Disease at the Istituto Zooprofilattico Sperimentale delle Venezie (IZSVe), Legnaro, Italy for confirmation and further genetic and molecular characterization.

Following the detection in poultry, an epidemiological investigation was carried out and Protection and Surveillance Zones were established according to Annex V of the Commission Delegated Regulation (EU) 2020/687. Five and 18 registered rural farms were present within the Protection (3 km) and Surveillance (10 km) Zones, respectively. In the latter zone, three commercial poultry farms were also present. All farms were visited by the LHU at least once a week, placing them under passive surveillance until the outbreak was extinguished and biological samples were to be taken, in case of suspicious signs or mortality, and any death or clinical signs of HPAI diseases were reported. 

As the LHU wanted to proceed with the culling of remaining birds and the destruc-tion of all carcasses for the rapid application of the biosecurity measures and mitigation of local exposure risk, on 7 November (five days after AIV outbreak confirmation) blood samples and cloacal swabs from two geese and one hen were collected for virological and serological investigations. No death or suspicious clinical signs were reported in this group of animals before the detection of the index case in the laying hens. In fact, these animals remained clinically healthy until the 11 November, when the LHU proceeded with the immediate authorization of the culling of these birds. In addition, another 63 healthy mixed breed domestic birds of different species (hens, ducks and geese) reared in two rural neighboring farmhouses considered at risk of lateral spread were culled. In the affected farm, 250 pigs, consisting of 170 adults and 80 piglets, were reared in the open, in an area separated from the infected laying hens’ holdings just by light fences, hence considered at risk of exposure.

In fact, the rural farm did not have adequate biosecurity measures in place. In particular, the various breeding sectors (see Figure 1) were outdoors and separated only by light fences allowing direct contact among the different species. Additionally, the structural features of the farm did not present any obstacle for direct contact with potentially infected wild birds and their fecal droppings.

Considering the possible epidemiological role of pigs as a “mixing vessel” species for avian, animal and human influenza viruses, an explorative serological investigation was carried out on 11 November to assess in a timely manner the possible exposure of pigs to AIV. Blood specimens were collected by convenience sampling from 67 clinically healthy pigs. In a second phase, upon evidence of serological positivity in the herd, individual nasal swabs from 174 pigs were taken to verify active virus circulation.

In compliance with the provisions of the National Legislation and in particular with that of the National Contingency Plan for Epidemic Emergencies relative to Avian Flu [18], since the representative sampling carried out on pigs exceeded that requested by the Manual consisting of a minimum 60 nasal swabs on removal of the infected birds and 60 blood samples two to four weeks after the first sampling of pigs (*n* = 67), and in view of the results obtained relative to the testing in these animals, these were allowed to move only to the slaughterhouse, starting from day 14 after the culling of the birds and completion of the cleaning and disinfection procedures.

## 3. Materials and Methods

### 3.1. Molecular Tests

Virological analyses consisted in adding 0.3 cm^3^ of organs (bowel, spleen, lung and air sac) collected from the laying hens and 0.5 mL of sterile isotonic PBS, pH 7.0–7.40, into a sterile tube containing a stainless-steel bead for homogenization at 30 Hz (Hertz) using a TissueLyser II (Qiagen, Hilden, Germany). The organ suspensions were centrifuged and supernatants were harvested. Total RNA was extracted from the supernatants using the QIAamp Viral RNA Mini Kit (Qiagen, Hilden, Germany) and tested by RRT-PCR for Influenza A matrix gene [19] for phylotyping, for H5 [20], and for the subtyping of the neuraminidase [21].

Cloacal swabs collected from two geese and 174 individual nasal swabs collected from apparently healthy pigs were placed in tubes containing transport medium for virological tests (1 mL of sterile isotonic PBS, pH 7.0–7.40). For each sample, 500 µL of the transport medium were added at a 1:1 ratio to a tissue lysis buffer for the purification of nucleic acids (ATL buffer, Qiagen, Hilden, Germany). After a centrifugation at 5000 g, a volume of 400 µL of the supernatant from each sample was used for RNA/DNA extraction with the automatic extractor QIAsymphony (DSP Virus/Pathogen Kit) (Qiagen, Hilden, Germany) following manufacturer’s instructions. The presence of AI virus in these swabs was assessed using a RRT-PCR targeting a conserved part of the matrix (M) gene according to the methods described by Spackman et al. [22] and Heine et al. [19]. Moreover, all samples were also tested for the presence of the endogenous β-actin gene, as an internal control [23] for extraction and amplification. 

### 3.2. Genome Amplification and Sequencing

Complete genome has been amplified using the SuperScript III One-Step RT-PCR System and Platinum Taq High Fidelity (Invitrogen, Carlsbad, CA, USA) using one pair of primers complementary to the conserved elements of the influenza A virus promoter as previously described [24]. The sequencing library was prepared by using the Nex-tera DNA XT Sample preparation kit (Illumina, San Diego, CA, USA) and quantified by using the Qubit dsDNA High Sensitivity Kit (Invitrogen, Carlsbad, CA, USA). The High Sensitivity DNA Analysis Kit (Agilent Technologies, Alpharetta, GA, USA) was used to determine average fragment length. The indexed libraries were pooled in equimolar concentrations and sequenced in multiplex for 300 bp paired-end on Illumina MiSeq, according to the manufacturer’s instructions.

### 3.3. Illumina Sequencing Data Analysis

The use of FastQC version 0.11.2 (https://www.bioinformatics.babraham.ac.uk/projects/fastqc/ (accessed on 6 March 2023)) was relied upon to evaluate the reading quality. Raw data were filtered by removal of reads with >10% of undetermined bases, while reads with >100 bases with a Q score of 80 bases were aligned against a reference genome by using BWA version 0.7.12 [25]. Picard-tools version 2.1.0 (http://picard.sourceforge.net (accessed on 6 March 2023)) and GATK version 3.5 [26] were used to correct potential errors, realign reads around indels, and recalibrate base quality. LoFreq version 2.1.2 [27] was used to call single-nucleotide polymorphisms. Outputs were used to generate consensus sequences.

### 3.4. Phylogenetic Analyses

Consensus sequences of the eight gene segments of the virus were compared with the most related sequences available in GISAID (https://www.gisaid.org/ (accessed on 6 March 2023)) and aligned by using MAFFT version 7 [28] IQTREE version 1.6.6 was used to construct the maximum likelihood phylogenetic trees performing ultrafast bootstrap resampling analysis (1000 replications) [29,30]. The phylogenetic tree was visualized using the FigTree version 1.4.2 (http://tree.bio.ed.ac.uk/software/figtree/ (accessed on 6 March 2023)).

### 3.5. ELISA Test

Blood from sixty-seven apparently healthy domestic pigs, two geese and one hen were collected, kept at room temperature until the serum separated from the clot. The serum was then transferred into tubes and stored at −20 °C. Sera were analysed by the enzyme-linked immunosorbent assay (ELISA), according to manufacturer’s instructions, using ID Screen^®^ Influenza A Anti-body Competition Multi Species (IDVet, Grabels, France) for detection of antibodies directed against the AIV nucleoprotein. The specificity of the test is reported by the manufacturer to be 100% (CI95: 97.7–100%) with a repeatability expressed by a Coefficient of Variation (CV) between 6% and 9%.

### 3.6. Hemagglutination Inhibition Tests

ELISA-positive sera were further analysed by the hemagglutination inhibition (HI) test performed according to the WOAH Terrestrial manual [31], using a panel of swine influenza virus (swIAV) and AIV antigens to determine subtype specific antibody titers. The H5N1 A/Etested by all the HI, MN and NI testsurasian wigeon/Italy/2020 (ew/IT) virus, a strain belonging to the 2.3.4.4b clade, was used as an antigenic surrogate to the H5N1 virus responsible for the outbreak. Furthermore, to exclude the possibility of a serological cross-reactivity between antibodies elicited by circulating swine influenza viruses and the chosen H5N1 antigen, sera were also analysed using the avian-like swIAV A/swine/Italy/6831-1/2013 (sw/IT/6831) of the H1N1 subtype belonging to clade HA-1C, the pandemic H1N1 virus A/Verona/Italy/2810/2009 (Italy/2810) of clade HA-1A, the H1N2 A/swine/Italy/166853-62/2013 (sw/IT/166853) of the human-like type belonging to clade HA-1B, and the H3N2 A/swine/Italy/8088-5/2010 (sw/IT/8088) of the human-like type, as representative antigens of the major enzootic strains in Europe [32,33]. Antigens of avian and swine origin derived from isolates obtained from embryonated chicken eggs or MDCK cells, respectively. ELISA and HI tests were first performed by Virology Unit of the Experimental Zooprophylactic Institute (IZS) of Lazio and Tuscany, and subsequently confirmed by the National Reference Center (CRN) at the IZSVe. Briefly, for the testing of swine sera, three volumes of receptor-destroying enzyme (RDE Seiken) were added to one volume of serum. The mixture was incubated overnight at 37 °C, and subsequently inactivated at 56 °C for 30 min and brought to a final dilution of 1:10 adding six volumes of PBS. To remove non-specific hemagglutinating factors, one volume of chicken erythrocytes were added to 10 volumes of serum and incubated at 4 °C under gentle shaking for 1 h, before removing erythrocytes by centrifugation at 2000 rpm for 10 min. HI tests were performed using four hemagglutinating units of virus with 0.5% chicken erythrocytes according to standard procedures. Two-fold dilutions were tested starting from 1:10 dilution. For the analysis of goose and laying hens sera, the protocol indicated by the WOAH was performed preincubating sera with packed chicken red blood cells to remove non-specific agglutination.

### 3.7. Neuraminidase Inhibition Assay

For a subset of high-titer H5 (e.g., equal or above 1:160) and ELISA positive swine sera we carried out the neuraminidase inhibition (NI) assay to identify antibodies against the N1 subtype. The NI assay was conducted using the H5N1 antigen ew/IT. In brief, sera were incubated for 20 min at room temperature (RT) in a 1:1 ratio with the test antigen previously diluted in a 1/20 dilution in a phosphate buffered solution. Fetuin was added and gently mixed before an overnight incubation at 37 °C. To convert free N-acetyl neuraminic acid into β-formyl pyruvic acid we added sodium periodate and incubated the solution for 30 min at 37 °C. The revelation of free β-formyl pyruvic was achieved upon addition of 2% sodium arsenite and thiobarbituric acid. In the absence of antibodies inhibiting neuraminidase enzymatic activity, a pink color was observed upon incubation at boiling temperature in a waterbath.

### 3.8. Microneutralization Assay

For a subset of high-titer H5 (e.g., equal or above 1:160) and ELISA positive swine sera, the microneutralization (MN) assay was carried out as a confirmatory test. The MN assay was performed according to the standard procedures described by the Center for Disease Control (CDC) [34]. 

Briefly, sera underwent the same RDE-based pretreatment adopted for HI testing. Two-fold dilutions of the treated sera were prepared in Dulbecco modified Eagle medium (D-MEM). The dilutions were then mixed to a 1:1 ratio with a virus solution containing 100 50% Tissue Culture Infectious Dose (TCID_50_) of the H5N1 A/urasian wigeon/Italy/2020 and incubated for 1 h at 37 °C. To each well, 1.5 × 10^4^ MDCK cells were added and incubated with the virus-serum mixture for 18–22 h, at 37 °C with 5% CO_2_. 

After incubation, cells were fixed with a cold solution of 80% acetone and incubated for 1 h at RT with a 1:1 mixture of primary anti-influenza A nucleoprotein monoclonal antibodies, MAB8257 and MAB8258 (Merckmillipore, Temecula, CA, USA), at a dilution of 1:4000 in blocking buffer. After washing, cells were incubated for 1 h with a secondary goat anti-mouse IgG antibody conjugated with horseradish peroxidase (HRP) at a dilution of 1:6000. After washing, a substrate based on o-phenylenediamine dihydrochloride (OPD) and citrate buffer was added to each well for 15 min at RT and then stopped with a 0.5 N sulfuric acid solution. 

The optical absorbance of wells was read at 490 nm and calculations were made to identify the reciprocal of the highest serum dilution resulting in 50% virus infection of cells.

## 4. Results

### 4.1. Virus Identification

The A/chicken/Italy/IZSLT-122448_21VIR9218-1/2021 virus detected in the laying hens was characterized as a HPAI H5N1 subtype strain. Whole-genome sequencing was performed and sequences were deposited to GISAID (EPI_ISL_7733644).

Phylogenetic analyses of the complete hemagglutinin (HA) gene showed that the virus belonged to the 2.3.4.4b clade and clustered with HPAI H5 viruses identified in Italy and Europe during 2021–2022 (Figure 2). Molecular analyses of the HA gene segment identified the mutation S137A (H3 numbering), which was described as being associated with increased binding to human-type receptors [35]; nevertheless, all the European HPAI H5Nx subtype viruses collected in Europe since 2020 possess this molecular feature.

The oro-nasal swabs collected from the pigs resulted RRT-PCR negative for the influenza type A matrix (M) gene by both applied protocols.

### 4.2. Serological Analyses

The two goose sera examined were found positive by ELISA and the HI test, recording titers of 128 and 2048 against the H5N1 A/urasian wigeon/Italy/2020 virus belonging to clade 2.3.4.4b, while serum from one hen tested positive by ELISA but negative by the HI test. Swine sera resulted to be positive by ELISA in 49/67 (73%) of the tested animals (Appendix A).

The HI test conducted on the 49 ELISA-positive animals identified 30/49 (61%) positive sera, with titres ranging from 1:10 to 1:1280 and a geometric mean equal to 1:64 ± SD 1:20 against the H5N1 A/urasian wigeon/Italy/2020 antigen (Figure 3). For seven of these sera, we observed values ranging from 1:320 to 1:1280. 

The HI tests carried out using three subtypes of swine influenza viruses and a pandemic H1N1 human influenza strain proved that 10/49 (20%) sera had detectable antibodies only against the H1N2 A/swine/Italy/166853-62 /2013 virus of the human-like type, recording a geometric mean value of 1:48 ± SD 1:20. Twenty-one of the 30 H5N1-positive sera tested negative by the HI test for the H1N2 A/swine/Italy/166853-62 /2013 virus. 

Microneutralization test (MN) was conducted using the H5N1 virus homologous to the one adopted for the HI assay. We tested by MN the H5N1-positive sera, selecting 9/30 samples with HI values greater than or equal to 1:160 with a GMT of 1:403 ± SD 1:333. The MN test recorded an overall high neutralizing ability of these sera ranging from 1:160 to 1:640, with a GMT of 1:359 ± SD 1:239.

The NI assay was conducted on the same nine high-titer sera and confirmed the presence of antibodies directed against the N1 protein of the ew/IT virus in all of the tested serum samples.

## 5. Discussion and Conclusions

In this report, we describe a multi-species outbreak caused by a 2.3.4.4b clade H5N1 HPAI virus in a rural farm in the province of Rome, Italy. The virus was detected in laying hens severely affected by the infection accompanied by high mortality, while geese, ducks and pigs in the same farm, although separated by light fences, remained healthy throughout the investigation. We could not detect viral genome in the nasal swabs collected from any of the sampled pigs, but we observed a robust seroconversion of the herd and in the few tested geese to an H5N1 HPAI virus homologous to the strain detected in the laying hens. 

To investigate the validity of this result, some of the HI positive swine sera were tested by a microneutralization assay that confirmed the presence of high titers of anti-H5 antibodies. Moreover, to assess the specificity of this response, we also tested the sera by the HI assay against influenza viruses circulating in pigs in Italy and we observed that the majority of the H5N1-positive sera were negative for all of the swine influenza antigens. Nonetheless, in several animals we recorded the presence of anti-NP antibodies in the absence of hemagglutination inhibition with all of the antigens tested. To further characterize the subtype of influenza virus that triggered this response, we performed the NI assay against the same H5N1 antigen used for the HI and MN assays and confirmed that high-titer sera carried antibodies inhibiting the enzymatic activity of the N1 neuraminidase.

The absence of clinical signs and virus replication in the nasal cavity, together with a variable HI response are all features compatible with a 2.3.4.4 clade virus infection in this species, according to data described by Kaplan et al. [36] in an experimental study, in which pigs were inoculated with 10^6^ EID_50_ of a North American 2.3.4.4 clade HPAI H5N8 virus. These animals did not present any sign of disease and developed an anti-NP response, but seroconversion by the HI assay was unreliable; moreover, nasal swabs resulted negative on days 1, 3 and 5 p.i. and replication of the virus was limited to the lower respiratory tract, failing to infect in contact sentinels.

The detection of 2.3.4.4b clade H5N6 viruses in healthy pigs in China [37]. together with a report of a specific seroconversion in a rural pig in France during the 2016–2017 HPAI epidemic [12]. provide field evidence in keeping with what reported in our investigation.

Nonetheless, we still lack experimental data in pigs on the fitness and virulence of recent 2.3.4.4b viruses circulating in Europe. Relevant genetic and ecological changes distinguish 2.3.4.4c and 2.3.4.4b viruses from previous epidemics in the United States and Europe from the 2.3.4.4b viruses affecting Europe in 2021/22. Therefore, our interpretation of the infection dynamics in this outbreak in light of previous experimental and epidemiological work has intrinsic limitations. 

Interestingly, the observed phenotype of infection appears to be milder than the ones described in previous experimental work based on other goose/Guangdong (gs/GD) lineage viruses of clade 0, 1, 2.1, 2.2 and 2.3, which induced moderate clinical manifestation, reaching low to moderate infectious titers at the level of the nose [15,38,39]. Irrespective of the genetic clade used for the challenge, all of these studies consistently reported lack of transmission to contact animals, indicating a low adaptation of the gs/GD lineage in this species. In light of this, we speculate that the high incidence of seroconversion observed among pigs during this outbreak is unlikely to be the result of secondary spread within the herd, rather the consequence of a prolonged exposure of the animals to high viral loads through fomites, contaminated soil and consumption of contaminated feed. In light of this, we hypothesize that the seroconversion observed among pigs during this outbreak is probably the result of secondary spread within the herd, via high contaminated fomites, soil or feed from infected birds.

Unfortunately, the data at our disposal were not sufficient to reach a conclusion on the intra-farm transmission chain.

Similar ecological and pathogenic drivers might have underpinned the HPAI H5N1 infections and seropositivity described in commercial and rural swine in Nigeria [14] China [40] and Indonesia [41], where endemicity for HPAI viruses in poultry favored spillover events to domestic mammals. Interestingly, in one of these studies, Meseko et al. [14] found HI titers against a 2.3.2.1c clade H5N1 virus comparable to the ones reported in our study, despite recording in healthy animals only high Ct values at the level of the trachea in an abattoir. Our investigation had several limitations. Virological sampling in pigs only took place about a week after the serological sampling, and about 17 days after the virus had led to almost 100% mortality in the flock of laying hens. Assuming that no secondary transmission occurred among pigs, the diagnostic sensitivity of our virological investigation was probably impaired by the delayed sampling. Moreover, for the logistical complexity inherent to the free-range setting of the farm, only adult pigs were sampled for blood, while piglets were not evaluated. Additionally, our approach to differential serology might have suffered from suboptimal sensitivity, due to the fact that the swine influenza antigens we used were not the most recent. This might explain in part the negative HI results for both avian and swine influenza in 19 pigs that had anti-NP antibodies. Due to our inability to detect and sequence the virus in pigs, we could not formulate a complete risk assessment, since molecular analyses necessary to identify markers of adaptation and virulence were only based on the virus identified in poultry in the same premise. Nonetheless, based on the negative rRT-PCR results on nasal swabs, we consider unlikely the possibility that a mammalian well-adapted reassortant strain between a swine influenza virus and the H5N1 virus was generated. The absence of symptoms and mortality in pigs in this and other studies is in stark contrast with the frequent reports of neurological syndromes associated with infections in wild mammals in Europe and North America, where 2.3.4.4b viruses have reached unprecedented levels of circulation [42].

This discrepancy might depend on the different anatomical route of infection, viral load, host susceptibility and the specific genotype responsible for the outbreak. Active serosurveillance in wildlife and farmed pigs will be key to understand the susceptibility of these species to infection with 2.3.4.4b H5Nx viruses. The reassortment of swine and H5Nx viruses of the Gs/GD lineage was previously reported [43] and poses a considerable risk for the emergence of zoonotic and prepandemic viruses, as demonstrated by the efficient aerosol transmission in guinea-pigs of hybrid HPAI H5N1 viruses carrying human-like polymerase and non-structural proteins [44].

In conclusion, we advocate for additional monitoring of swine in close proximity or exposed to HPAI infected poultry, to achieve a timely identification of spillover events, given the delicate role played by this species in the emergence of reassortant zoonotic viruses. Functional separation of poultry species and mammals should be prioritized as a preventive measure to reduce the risk of inter-species transmission.

## Figures and Tables

**Figure 1 microorganisms-11-01162-f001:**
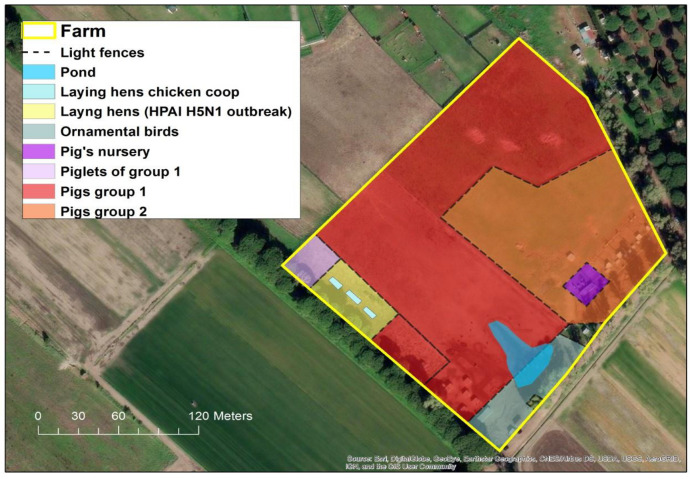
Location of breeding areas for the various animal species bred in the HPAI H5N1 outbreak. (Software GIS used: ESRI 2020. ArcGIS Desktop: Release 10.8.1 Redlands, CA, USA: Environmental Systems Research Institute. Background used: World Imagery (Esri, DigitalGlobe, GeoEye, i-cubed, USDA FSA, USGS, AEX, Getmapping, Aerogrid, IGN, IGP, swisstopo, and the GIS User Community). Spatial resolution of the map: 300 dpi).

**Figure 2 microorganisms-11-01162-f002:**
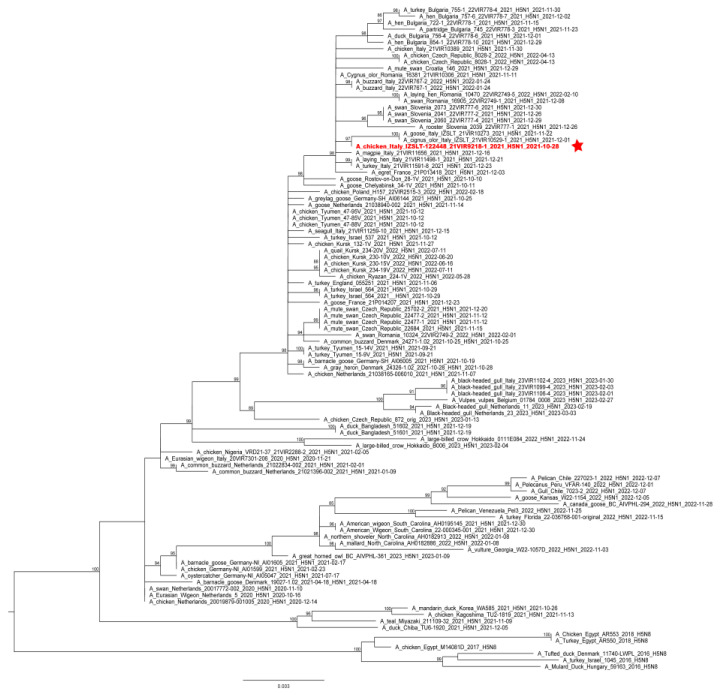
ML phylogenetic tree for the HA gene segment of H5NX avian influenza viruses from the 2.3.4.4b clade. The sequence of the virus causing the outbreak is highlighted in red and sided by a red star. The numbers at the nodes represent bootstrap values.

**Figure 3 microorganisms-11-01162-f003:**
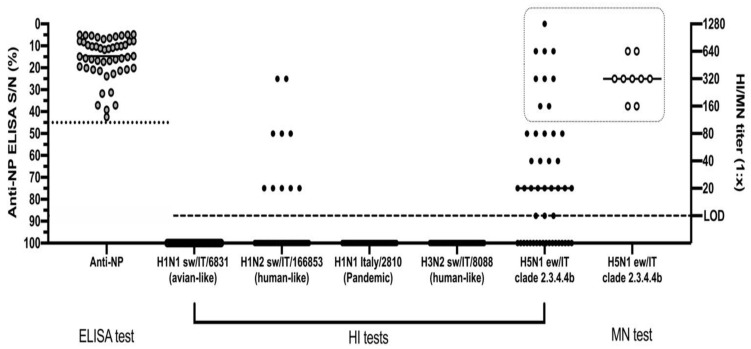
Serological screening of anti-NP ELISA positive pigs by HI and MN assays. The dotted line represents the threshold of S/N positivity of the ELISA test. The dashed line represents the limit of detection of the HI and MN tests. The dotted square highlights the samples tested by all of the HI, MN and NI tests against the H5N1 virus. Data are presented as individual and mean values for each type of analysis.

## Data Availability

The data presented in this study are available on request from the corresponding author.

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
