# Peer review of "Seroconversion of a Swine Herd in a Free-Range Rural Multi-Species Farm against HPAI H5N1 2.3.4.4b Clade Virus"

_microorganisms, 2023, doi:10.3390/microorganisms11051162_

Round 1
Reviewer 1 Report
HPAIV H5N1 clade 2.3.4.4b has exacerbated since early 2022 into a panzootic. Possible adaption of AIV to mammalian livestock hosts and subsequent human exposure is of particular concern. This manuscript describes the seroconversion of pigs in an Italian multi-species farm against HPAI H5N1 clade 2.3.4.4b virus, suggesting it would have been infected naturally after close contact with HPAI-H5N1 contaminated poultry.
1. Please delete the minus symbol “-“ in the middle of words throughout the manuscript (e.g. lines 42, 68, 201, 272, 379).
2. Line 49: The authors state that over “XX species of predators” were found. Please specifiy the number of species.
3. Line 93: the abbreviation for avian influenza viruses is (like written in line 69) “AIV” – not IAV (which stands for Influenza A virus)
4. Line 109 and line 152: Why did you only test a few birds (3 laying hens + 2 geese)?
5. Line 122: “periodically” - how many intervals of sampling for how long/ is there a time scheme?
6. Was there any former AIV and/or swIAV outbreak reported in this farm, including the infection of the pigs?
7. Line 145: please use superscript letters for cm3 specification.
8. Line 233: same here, please use superscript letters for TCID50.
9. Authors should use a consistent writing style for numerals (either spelled out or written as numerals up to number 10)
10. Line 182: please add a dot at the end of the sentence (same in line 189).
11. Line 202: the authors should consider switching to the more common abbreviation "swIAV" instead of "SIV".
12. Line 204: please delete one “to the” – this is doubled here.
13. Please provide more information on the antigen of the outbreak used for antibody detection in lines 203-205. Was it a cell culture or chicken egg derived isolate?
14. One minor weakness is the poor documentation of the biosecurity data on this multi-species farm.
Author Response
Dea Editor, attached you will find the answers to the comments of Revewer 1.

Reviewer 2 Report
This paper describes the transmission of H5 subtype high pathogenicity avian influenza virus (HPAIV) to pigs on a free-range farm in Italy in 2021. The detection of specific antibodies in pigs kept on the same farm, which suggests the evidence of infection with an H5 HPAIV from poultry to pigs, is an important finding for the future countermeasures against the infection of HPAIV not only in poultry but also in other mammalian domestic animals.
<Major comments>
1. Title.
The neuraminidase inhibition (NI) test must be performed in addition to the hemagglutination inhibition (HI) and micro-neutralization (MN) tests, since the authors have just determined antibodies again H5 HA antigen. This will truly prove that pigs have antibodies to the virus of subtype H5N1, if you can detect specific antibodies against H5 HA and N1 NA, respectively.
2. Fig. 1
Viruses of clade 2.4.4.4b have been isolated outside Europe, in North America, and in Asian regions. These viruses should be included in the phylogenetic tree. If subgroups of the isolates were classified for viruses isolated in the winter of 2021-2022, they should be visualized in this figure. Additionally, there are several places where bootstrap values overlap with the phylogenetic tree, which should be corrected.
3. Results or discussion
Were the pigs from this farm eventually culled? Or were the pigs allowed to move after a certain quarantine period? Information should be added as to what government action was taken.
<Minor comments>
L145:cm3 >>superscript
L190: ELISA, not Elisa
L193: “en dash” –20oC, not -20oC
L212: ELISA
L217: 30 min
L233: TCID50 >>subscript
L234: 104 MDCK >>superscript
L308: antibodies, not anti-bodies
L313: 106 EID50 >>superscript, subscript
L359: delete “line break”
L379: exposed, not ex-posed
Author Response
Dear Editor, attached you will find the answers to the comments of Reviewer 2, and the Article with uor changes and addition in "track change "modality.
Best regards
